# Examining the Double Burden of Underweight, Overweight/Obesity and Iron Deficiency among Young Children in a Canadian Primary Care Setting

**DOI:** 10.3390/nu15163635

**Published:** 2023-08-18

**Authors:** Sean A. Borkhoff, Patricia C. Parkin, Catherine S. Birken, Jonathon L. Maguire, Colin Macarthur, Cornelia M. Borkhoff

**Affiliations:** 1Division of Pediatric Medicine and the Pediatric Outcomes Research Team (PORT), Hospital for Sick Children, Toronto, ON M5G 1E8, Canada; saborkhoff@uwaterloo.ca (S.A.B.); patricia.parkin@sickkids.ca (P.C.P.); catherine.birken@sickkids.ca (C.S.B.); colin.macarthur@sickkids.ca (C.M.); 2Institute of Health Policy, Management and Evaluation, University of Toronto, Toronto, ON M5T 3M6, Canada; jonathon.maguire@utoronto.ca; 3Child Health Evaluative Sciences, SickKids Research Institute, Toronto, ON M5G 1X8, Canada; 4Department of Pediatrics, Faculty of Medicine, University of Toronto, Toronto, ON M5S 1A8, Canada; 5Li Ka Shing Knowledge Institute, St. Michael’s Hospital, Toronto, ON M5B 1A6, Canada

**Keywords:** underweight, overweight/obesity, iron deficiency, double burden of malnutrition, early childhood, primary care

## Abstract

There is little evidence on the prevalence of the double burden and association between body mass index (BMI) and iron deficiency among young children living in high-income countries. We conducted a cross-sectional study of healthy children, 12–29 months of age, recruited during health supervision visits in Toronto, Canada, and concurrently measured BMI and serum ferritin. The prevalence of a double burden of underweight (zBMI < −2) and iron deficiency or overweight/obesity (zBMI > 2) and iron deficiency was calculated. Regression models examined BMI and serum ferritin as continuous and categorical variables, adjusted for covariates. We found the following in terms of prevalence among 1953 children (mean age 18.3 months): underweight 2.6%, overweight/obesity 4.9%, iron deficiency 13.8%, iron-deficiency anemia 5.4%, underweight and iron deficiency 0.4%, overweight/obesity and iron deficiency 1.0%. The change in median serum ferritin for each unit of zBMI was −1.31 µg/L (95% CI −1.93, −0.68, *p* < 0.001). Compared with normal weight, we found no association between underweight and iron deficiency; meanwhile, overweight/obesity was associated with a higher odds of iron deficiency (OR 2.15, 95% CI 1.22, 3.78, *p* = 0.008). A double burden of overweight/obesity and iron deficiency occurs in about 1.0% of young children in this high-income setting. For risk stratification and targeted screening in young children, overweight/obesity should be added to the list of important risk factors.

## 1. Introduction

In a 2017 policy brief, the World Health Organization (WHO) raised concerns about the double burden of malnutrition (DBM), defined as “the coexistence of undernutrition along with overweight, obesity or diet-related noncommunicable diseases, within individuals, households and populations, and across the life-course” [1]. The WHO included examples of individual-level DBM (simultaneous development of two of more types of malnutrition, e.g., obesity with nutritional anemia); household-level DBM (e.g., nutritional anemia in a mother and her child with overweight); and population-level DBM (e.g., both undernutrition and overweight/obesity coexisting in the same region or nation). While the WHO describes DBM as a global challenge, the 2020 Lancet series notes that DBM affects mostly low-income and middle-income countries (LMICs), especially those undergoing rapid nutrition transition [1,2].

In a recent systematic review aimed at identifying published operational definitions of DBM, Davis et al. reported on 239 articles (1992–2017) and proposed a framework including: level of assessment (population, household, individual), target population (age, sex), and forms of malnutrition (undernutrition, overnutrition) [3]. The authors found that studies of DBM prevalence predominately reported the coexistence of underweight and overweight/obesity at the population level in mixed target populations living in LMICs; far fewer studies reported on other forms of malnutrition, such as micronutrient deficiencies (e.g., iron deficiency), and coexistence at the individual level among those living in high-income countries. Our study aims to address this gap.

In high-income countries such as the United States and Canada, both growth monitoring and assessment of risk for iron deficiency are recommended for all preschool-aged children at each health supervision visit in primary care [4,5,6,7]. This provides an ideal setting to identify the coexistence (double burden) of suboptimal growth and iron deficiency at the individual level. However, few studies have examined the association between underweight, overweight/obesity and iron deficiency, and suboptimal growth is not included as a risk factor for screening children for iron deficiency [5,7,8,9,10].

We hypothesized that young children with suboptimal growth (underweight and overweight/obesity) would be at risk of iron deficiency. The objective of this study was to examine the prevalence of individual-level DBM, defined as the coexistence of underweight or overweight/obesity and iron deficiency among young children in a Canadian primary care setting. We also examined the association between body mass index (as a continuous BMI z score and body weight categories) and iron deficiency in early childhood. 

## 2. Materials and Methods

### 2.1. Study Design

We used a cross-sectional study design. Consent was obtained from parents and ethics approval was obtained from the Hospital for Sick Children and St. Michael’s Hospital. We followed the Strengthening the Reporting of Observational Studies in Epidemiology (STROBE) reporting guideline for observational cohort studies [11].

### 2.2. Study Setting and Population

Study participants were recruited by research personnel during scheduled health supervision visits at nine primary care practices participating The Applied Research Group for Kids (TARGet Kids!) practice-based research network between December 2008 and March 2020. TARGet Kids! is an ongoing open longitudinal cohort focusing on early-life exposures set in Toronto, Canada. The cohort profile has been previously described [12]. Parents of young children completed a standardized questionnaire collecting sociodemographic and nutritional information, and trained research personnel obtained blood samples and anthropometric measurements. 

Children were included if they were attending a scheduled 12-, 15-, 18-, or 24-month health supervision visit (age range 12–29 months) and had a blood sample. TARGet Kids! cohort exclusion criteria are as follows: children with health conditions that affect growth (e.g., failure to thrive, cystic fibrosis); chronic conditions (except asthma); severe developmental delay; and parents who cannot communicate in English. Additional exclusion criteria specific to this study were children with the following: gestational age < 32 weeks; receiving iron supplementation; missing body mass index (BMI) data; serum ferritin > 200 ug/L (beyond the upper limit of the reference interval); and C-reactive protein (CRP) > 5 mg/L or missing and serum ferritin > 12 ug/L (elevated CRP may indicate acute inflammation and falsely elevate serum ferritin) [13].

### 2.3. Exposure Variable

The exposure variable was child age- and sex-standardized body mass index z-score (zBMI). Trained research assistants embedded within each practice site obtained growth measures following standardized WHO procedures [14]. For children < 2 years, weight was measured using an infant scale and length using a length board. For children ≥ 2 years, weight was measured using a calibrated precision digital scale and standing height using a stadiometer. BMI (kg/m^2^) was calculated by dividing the child’s weight in kilograms by their length/height in meters squared. Body mass index was evaluated as both a continuous variable (zBMI) and a categorical variable with body weight categories defined as follows: underweight (z < −2), normal weight (−2 ≤ z ≤ 1), at-risk-of-overweight (1 < z ≤ 2), and overweight/obese (z > 2), as recommended for children from birth to five years [15,16,17].

Although weight-for-length has been recommended by some investigators for children less than 2 years of age, recent studies have found high agreement between weight-for-length and BMI-for-age, which was the exposure variable used in our study [18,19].

### 2.4. Outcome Variable 

The outcome variable was iron status, as measured by serum ferritin. We also measured hemoglobin and CRP. Research assistants collected blood samples, which were refrigerated at the practice sites and transported the same day to the laboratory (www.mountsinaiservices.com (accessed on 22 July 2023)). Serum ferritin and CRP were analyzed on the Roche platform (Basel, Switzerland); hemoglobin was analyzed on the Sysmex XN-9000 Hematology Analyzer (Kobe, Japan).

Iron status was evaluated as both a continuous and binary variable. The continuous measure of iron status was serum ferritin in µg/L. The binary measure (yes/no) of iron status was based on WHO guidelines: ID was defined as serum ferritin < 12 µg/L; IDA was defined as serum ferritin < 12 µg/L and hemoglobin ≤ 110 g/L [20]. CRP was dichotomized as ≤1.0 mg/L and >1.0 mg/L to ≤5.0 mg/L to account for children with low-grade systemic inflammation [13].

If a child had data from multiple visits, the first visit at which a blood sample was obtained was used for analysis. 

### 2.5. Other Variables 

Covariates collected through a parent-reported questionnaire were the following: child age, sex, birthweight, family income (response to the question: “What was your total family income before taxes last year?”), infant feeding in the first year of life (mostly breastfed, breast milk and formula equally, mostly formula-fed), breastfeeding duration ≥ 12 months, daily cow’s milk > 2 cups, bottle use > 15 months of age. CRP level (mg/L) was also included as a covariate [13].

### 2.6. Statistical Analysis 

We generated descriptive statistics for the entire sample and for each of the four weight categories: underweight, normal weight, at-risk-of-overweight, overweight/obese. We calculated the prevalence (and 95% confidence intervals, CIs) of underweight, at-risk-of-overweight, overweight/obese, and iron deficiency, as well as the double burden of underweight and iron deficiency or overweight/obese and iron deficiency.

We constructed four multivariable models. Linear regression models were used to examine the association between zBMI and the continuous variable serum ferritin (µg/L), and the association between each of the four weight categories and serum ferritin. Given the residuals were not normally distributed, the beta coefficients for serum ferritin values were log transformed to fit the linear regression model and then back transformed to allow the results to be interpreted. Results were expressed as change in median serum ferritin and 95% CI. Logistic regression models were used to examine the association between zBMI and iron deficiency (serum ferritin < 12 µg/L), and the association between each of the four weight categories and iron deficiency. Results were expressed as odds ratios (ORs) and 95% CIs. 

All models were adjusted for all covariates previously described regardless of statistical significance [21]. Restricted cubic spline transformation and then Loess curves were used to confirm the non-linear association between age and serum ferritin; 3 knot points were selected to correspond to the scheduled visits (15, 18 and 24 months) in the middle of the age range at outcome [22]. All potential confounders had <15% missing data, except self-reported family income, which had 18% missing data. Multiple imputation was used to impute missing covariate data [23]. To reduce the potential for bias, models were run on 20 imputed data sets [24]. Results of the 20 imputed data sets were combined, and the parameter estimates (95% CIs) for the adjusted pooled models were reported. Statistical significance was defined as *p* < 0.05; all statistical tests were 2-sided. Statistical analysis was conducted using SAS 9.4 statistical software (SAS Institute, Cary, NC, USA).

## 3. Results

There were 2575 healthy children between the ages 12 and 29 months with a blood sample enrolled in the TARGet Kids! cohort. Of these, 622 (24.2%) were excluded because of the following: gestational age < 32 weeks (*n* = 83); receiving iron supplementation (*n* = 187); missing BMI (*n* = 208); serum ferritin value > 200 ug/L (*n* = 1); and CRP value either >5 mg/L or missing and a serum ferritin value of >12 ug/L (*n* = 143). Children with and without missing data on BMI are comparable (Appendix A). The final sample included 1953 children (Table 1). Descriptive statistics for the response sample and final analytic sample (with imputation) are shown in Table 2, demonstrating minimal selection bias owing to missing data [25].

Table 3 provides the characteristics of study participants for the total sample and by weight category (quantified using zBMI). The mean age was 18.3 (SD 5.0) months and 945 (48.4%) were female. The total sample had a mean zBMI of 0.13 (SD 1.1) and a mean serum ferritin of 27.4 (SD 18.7) µg/L.

The prevalence of each form of malnutrition was as follows: underweight 2.6% (95% CI 1.9, 3.4; *n* = 51), at-risk-of-overweight 16.0% (95% CI 14.3, 17.9; *n* = 312), overweight/obesity 4.9% (95% CI 3.9, 5.9; *n* = 95), iron deficiency 13.8% (95% CI 12.2, 15.5; *n* = 269), and iron-deficiency anemia 5.4% (95% CI 4.3, 6.6; *n* = 92). The prevalence of double burden of malnutrition was as follows: underweight and iron deficiency 0.4% (95% CI 0.2, 0.8; *n* = 8), overweight/obesity and iron deficiency 1.0% (95% CI 0.6, 1.6; *n* = 20).

Table 4 and Table 5 show the association of zBMI and weight category with median serum ferritin and iron deficiency (serum ferritin < 12 µg/L).

In the multivariable linear regression analysis (serum ferritin as continuous variable), higher zBMI was associated with lower serum ferritin (change in median serum ferritin −1.31 µg/L, 95% CI −1.93, −0.68, *p* < 0.001). Compared with normal-weight children, there was no association between underweight and serum ferritin (1.95 µg/L, 95% CI −2.28, 6.97, *p* = 0.39) or at-risk-of-overweight and serum ferritin (−1.41 µg/L, 95% CI −3.14, 0.46, *p* = 0.14); meanwhile, overweight/obesity was associated with lower serum ferritin (−4.81 µg/L, 95% CI −7.27, −2.02, *p* = 0.001) (Table 4).

In the multivariable logistic regression analysis (iron deficiency as categorical variable), higher zBMI was associated with a higher odds of iron deficiency (OR 1.14, 95% CI 1.00, 1.30, *p* = 0.04). Compared with normal-weight children, there was no association between underweight and iron deficiency (OR 0.98, 95% CI 0.44, 2.23, *p* = 0.97) or between at-risk-of-overweight and iron deficiency (OR 1.16, 95% CI 0.80, 1.70, *p* = 0.44); meanwhile, overweight/obesity was associated with a higher odds of iron deficiency (OR 2.15, 95% CI 1.22, 3.78, *p* = 0.008) (Table 5). 

Covariates associated with lower serum ferritin and higher odds of iron deficiency were as follows: child age 12–15 months; family income < 40,000 CAD; breastfeeding duration ≥ 12 months; and daily cow’s milk intake > 2 cups. Covariates associated with higher serum ferritin and lower odds of iron deficiency were the following: higher birthweight; mostly formula-fed in the first year of life; breast milk and formula equally in the first year of life (Table 6 and Table 7).

## 4. Discussion

In 1953 healthy young children aged 12–29 months at scheduled health supervision visits in an urban Canadian primary care setting, we found a prevalence of underweight 2.6%, overweight/obesity 4.9%, iron deficiency 13.8%, and iron-deficiency anemia 5.4%. The prevalence of a double burden of malnutrition was as follows: underweight and iron deficiency 0.4%, and overweight/obesity and iron deficiency 1.0%.

We found that higher BMI was associated with lower serum ferritin and higher odds of iron deficiency, and overweight/obesity was associated with lower serum ferritin and higher odds of iron deficiency. We found no association between underweight and iron deficiency. Considering the DBM framework proposed by Davis et al. [3], these findings constitute individual-level coexistence (double burden) of overweight/obesity and iron deficiency in preschool children living in a high-income country. 

Few studies have examined the association of underweight or overweight/obesity with iron deficiency in young children living in a high-income setting. We previously identified a strong association between higher body mass index and iron deficiency in young children [9]. However, underweight children were excluded from our previous analysis. Similarly, Brotanek et al. examined data from the United States National Health and Nutrition Examination Survey IV (NHANES, 1999–2002) to identify risk factors for iron deficiency in 1641 children aged 1–3 years [10]. Children with overweight had a higher odds of iron deficiency (3.34 95% CI 1.10–10.12). However, the approach by Brotanek et al. differed from our approach. First, the referent group in Brotanek et al. included both normal-weight and underweight children, whereas our referent group included normal-weight children only. Second, Brotanek et al. defined iron deficiency according to any two of three abnormal laboratory tests (transferrin saturation, free erythrocyte protoporphyrin, serum ferritin), whereas we defined iron deficiency according to serum ferritin only. However, neither of these two previous studies from Canada and the U.S.A. examined the independent association of underweight with iron deficiency or examined the DBM. 

Guivarch et al. examined underweight at 1 year as one of several predictors of iron deficiency at 2 years in children (*n* = 568) recruited from primary care pediatricians’ practices in France [8]. The authors reported underweight at 1 year in 1.5% and iron deficiency at 2 years in 6.7%. In our cross-sectional cohort (*n* = 1953) with a mean age of 15.5 months for underweight children, we found underweight in 2.6% and iron deficiency in 13.8%. For the association between underweight and iron deficiency, Guivarch et al. found an adjusted OR of 5.9 with a wide confidence interval including 1.0 (95% CI 1.0, 34.0), whereas we found an adjusted OR of 0.98 with a narrower confidence interval crossing 1.0 (95% CI 0.44, 2.23). Guivarch et al. included low weight at 1 year in their four-item prediction model, while we have found no association between underweight and iron deficiency. Of note, Guivarch et al. did not examine overweight/obesity as a risk factor for iron deficiency.

In our current analysis, we report the prevalence of DBM for underweight and overweight/obesity with iron deficiency, which allows for comparisons with other countries. DBM has been examined in several large cohorts, most reporting on prevalence without assessment of association or risk. Findings vary depending on country, level of assessment (population or individual), and measures (for example, analysis of anemia measured with hemoglobin or iron deficiency measured with serum ferritin). 

Engle-Stone et al. examined data from population-based surveys of preschool children from 21 countries (only one high-income country: U.S.A.) from the Biomarkers Reflecting Inflammation and Nutritional Determinants of Anemia (BRINDA) project [16]. The authors reported on individual-level DBM and found that overweight/obesity (zBMI > 2) and iron deficiency were associated in only 3 of 23 surveys for which ferritin was measured; one of these was the survey from the U.S.A. (2006), where the prevalence of iron deficiency was 12.9% and coexistent overweight/obesity and iron deficiency was 2.2%. 

Varghese and Stein examined data from the Indian Fourth National Family Health Survey (NFHS-4, 2015–2016) [26]. The authors reported on the median state-level prevalence of nutritional burdens in children (*n* = 145,653) aged 6–59 months: underweight 18.1%, overweight 1.5%, anemia 57.3%, underweight and anemia 11.3%, overweight and anemia 0.8%. 

Castillo and Suarez-Ortegon examined data from the Colombian National Survey of Nutritional Situation Columbia (ENSIN, 2015) [27]. The authors reported on the individual-level DBM in children (*n* = 7434) aged 1–4 years: overweight (zBMI > 2) 5.3%, iron deficiency 14.9%, overweight and iron deficiency 1.4%.

Kamruzzaman examined data from the Bangladesh Demographic and Health Survey 6th round (BDHS, 2011) [28]. The author reported on individual-level DBM in children (*n* = 2373) aged 6–59 months: underweight 12.9%, overweight 1.7%, anemia 51.2%; however, when using multiple statistical approaches, the author found no relationship between BMI and anemia.

To address individual-level malnutrition, professional organizations such as the American Academy of Pediatrics (AAP) and the Canadian Paediatric Society (CPS) recommend both growth monitoring and assessment of risk for iron deficiency for all preschool-aged children at each health supervision visit in primary care [4,5,6,7]. These visits are scheduled frequently in the first 2 years of life, thus providing an ideal setting for early identification and intervention. The risk factors for iron deficiency listed by the AAP and CPS include the following: low socioeconomic status, prematurity or low birthweight, exposure to lead, exclusive breastfeeding beyond 4 months of age without supplemental iron, weaning to whole milk or complementary foods that are not rich in iron, prolonged bottle feeding, and those living with special healthcare needs or chronic disease. Our findings suggest that for otherwise healthy young children, overweight/obesity (zBMI > 2) should also be considered a risk factor for iron deficiency, while we did not find underweight (zBMI < −2) to be a risk factor. 

We included many covariates in our models; some were associated with lower serum ferritin and higher odds of iron deficiency (longer breastfeeding duration, daily cow’s milk intake > 2 cups, family income < 40,000 CAD), and others were associated with higher serum ferritin and lower odds of iron deficiency (higher birthweight, mostly formula-fed in the first year of life, breast milk and formula equally in the first year of life). In previous analyses, we have found several of these variables, and the risk factors listed by the AAP and CPS, to be independently associated with iron deficiency [29,30,31,32,33].

To examine the association between BMI and iron deficiency, we used serum ferritin and CRP. These laboratory tests are recommended by the AAP for confirming iron deficiency as they are readily available and interpretable in clinical practice [5]. There has been recent interest in the role of hepcidin, a peptide hormone that is predominantly produced in the liver but also in adipose tissue [34]. Hepcidin regulates dietary iron absorption and is increased in inflammatory states, including low-grade chronic inflammation due to obesity. In a sample of young children, aged 6 months to 3 years, the hepcidin level was found to be strongly correlated with serum ferritin and CRP [35]. Hepcidin has been proposed as a potential mediator of the association between obesity and iron deficiency [36].

Strengths of this study include the large sample size of healthy children attending scheduled health supervision visits in primary care. Both exposure (zBMI) and outcome (serum ferritin) were measured objectively. We also adjusted for several possible confounding variables in our analyses. 

Limitations of this study include the use of a cross-sectional design, which does not allow for the determination of causality. Furthermore, the participants were recruited from community pediatric practices in an urban Canadian setting and may not be representative of children of the same age living in different settings; however, the rates of underweight, overweight/obesity, and iron deficiency were similar to those noted in other high-income countries. We also adjusted for several social determinants.

## 5. Conclusions

Using a cross-sectional design, we found an association between higher body mass index and iron deficiency in young children aged 12–29 months at scheduled health supervision visits in primary care in a high-income setting. We found no association between underweight and iron deficiency, but did find an association between overweight/obesity and iron deficiency, representing individual-level DBM, which occurred in about 1.0%. There is a paucity of such studies in high-income countries. For risk stratification and targeted screening, overweight/obesity (zBMI > 2) should be added to the list of important risk factors.

## Figures and Tables

**Table 1 nutrients-15-03635-t001:** Participant recruitment and selection for inclusion in the cross-sectional cohort (*n* = 1953).

Characteristics	No.
Healthy children between ages 12 and 29 months with a blood sample enrolled in TARGet Kids! cohort	2575
Exclusion criteria	
Gestational age < 32 weeks	83
Receiving iron supplementation	187
Missing zBMI value	208
Serum ferritin value > 200 μg/L	1
CRP > 5 mg/L and SF ≥ 12 μg/L	134
CRP missing and SF ≥ 12 μg/L	9
Final cohort	1953

**Table 2 nutrients-15-03635-t002:** Characteristics of 1953 study participants.

Characteristics	Response Sample		Imputed Sample
	*n*	Mean (SD) or N (%)	Mean (SD) or N (%) ^1^
**Patient-level characteristics**			
Child age, months	1953	18.3 (5.0)	18.3 (5.0)
Child sex, female	1953	945 (48.4)	945 (48.4)
zBMI	1953	0.13 (1.1)	0.13 (1.1)
Weight category	1953		
Underweight (zBMI < −2)		51 (2.6)	51 (2.6)
Normal weight (−2 ≤ zBMI ≤ 1)		1495 (76.6)	1495 (76.6)
At-risk-of-overweight (1 < zBMI ≤ 2)		312 (16.0)	312 (16.0)
Overweight/obese (zBMI > 2)		95 (4.9)	95 (4.9)
Birthweight, kg	1835	3.3 (0.6)	3.3 (0.6)
Maternal ethnicity ^3^	1725		
European		1112 (64.5)	1255 (64.3)
Non-European		613 (35.5)	698 (35.7)
Maternal education	1763		
High school or less		141 (8.0)	154 (7.9)
College/university		1622 (92.0)	1799 (92.1)
Family income (CAD)	1604		
Less than 40,000 CAD		161 (10.0)	200 (10.2)
40,000–79,999 CAD		227 (14.2)	270 (13.8)
80,000–149,999 CAD		546 (34.0)	644 (33.0)
150,000+ CAD		670 (41.8)	839 (43.0)
Gestational age	1687		
32–36 weeks		187 (11.1)	210 (10.8)
≥37 weeks		1500 (88.9)	1743 (89.3)
**Infant feeding practices**			
Infant feeding in first year of life	1720		
Mostly breastfed		1086 (63.1)	1229 (62.9)
Breast milk and formula equally		327 (19.0)	371 (19.0)
Mostly formula-fed		307 (17.9)	353 (18.1)
Breastfeeding duration ≥ 12 months	1640	869 (53.0)	1015 (52.0)
Bottle use > 15 months	1612	480 (29.8)	624 (32.0)
Daily cow’s milk intake > 2 cups (500 mL)	1620	422 (26.1)	512 (26.2)
Meat consumption in last 3 days	1816	1754 (96.6)	1883 (96.4)
**Laboratory characteristics**			
Serum ferritin (µg/L)	1953	27.4 (18.7)	27.4 (18.7)
Hemoglobin (g/L) ^2^	1711	118.1 (8.7)	118.1 (8.7)
Iron deficiency	1953	269 (13.8)	269 (13.8)
Iron-deficiency anemia ^2^	1711	92 (5.4)	92 (5.4)
CRP (mg/L)	1953	0.6 (1.0)	0.6 (1.0)
≤1.0		1648 (84.4)	1648 (84.4)
>1.0 to ≤5.0		305 (15.6)	305 (15.6)

Data regarding baseline characteristics are presented as mean (SD) or N (%). zBMI indicates body mass index z score; CRP indicates C-reactive protein. ^1^ Based on the final analytic sample: data are for one imputed data set. ^2^ Based on the 1711 children with complete data for IDA. ^3^ Maternal ethnicity: European includes Western European, Eastern European, and Australian or New Zealander; Non-European includes East Asian, Southeast Asian, South Asian, West Asian, African, Caribbean, Latin American, North American Indigenous, and Mixed (2 or more ethnic groups).

**Table 3 nutrients-15-03635-t003:** Characteristics of study participants for the total sample and by weight category (quantified using zBMI) ^1^.

	Total	Underweight	Normal Weight	At-Risk-of- Overweight	Overweight /Obese
		zBMI < −2	−2 ≤ zBMI ≤ 1	1 < zBMI ≤ 2	zBMI > 2
N	1953	51 (2.6)	1495 (76.6)	312 (16.0)	95 (4.9)
**Patient-level characteristics**					
Child age, months	18.3 (5.0)	15.5 (4.4)	18.0 (5.0)	19.9 (4.7)	20.6 (4.5)
Child sex, female	945 (48.4)	24 (47.1)	735 (49.2)	142 (45.5)	44 (46.3)
zBMI	0.13 (1.1)	−2.44 (0.3)	−0.20 (0.7)	1.41 (0.3)	2.54 (0.5)
Birthweight, kg	3.3 (0.6)	3.1 (0.6)	3.3 (0.6)	3.4 (0.6)	3.5 (0.6)
Maternal ethnicity ^3^					
European	1255 (64.3)	27 (52.9)	950 (63.6)	214 (68.6)	64 (67.4)
Non-European	698 (35.7)	24 (47.1)	545 (36.5)	98 (31.4)	31 (32.6)
Maternal education					
High school or less	154 (7.9)	2 (3.9)	109 (7.3)	29 (9.3)	14 (14.7)
College/university	1799 (92.1)	49 (96.1)	1386 (92.7)	283 (90.7)	81 (85.3)
Family income (CAD)					
Less than 40,000 CAD	200 (10.2)	8 (15.7)	149 (10.0)	31 (9.9)	12 (12.6)
40,000–79,999 CAD	270 (13.8)	9 (17.7)	209 (14.0)	41 (13.1)	11 (11.6)
80,000–149,999 CAD	644 (33.0)	18 (35.3)	493 (33.0)	98 (31.4)	35 (36.8)
150,000+ CAD	839 (43.0)	16 (31.4)	644 (43.1)	142 (45.5)	37 (39.0)
Gestational age					
32–36 weeks	210 (10.8)	5 (9.8)	167 (11.2)	27 (8.7)	11 (11.6)
≥37 weeks	1743 (89.3)	46 (90.2)	1328 (88.8)	285 (91.4)	84 (88.4)
**Infant feeding practices**					
Infant feeding in first year of life					
Mostly breastfed	1229 (62.9)	37 (72.6)	968 (64.8)	173 (55.5)	51 (53.7)
Breast milk and formula equally	371 (19.0)	5 (9.8)	255 (17.1)	82 (26.3)	29 (30.5)
Mostly formula-fed	353 (18.1)	9 (17.7)	272 (18.2)	57 (18.3)	15 (15.8)
Breastfeeding duration ≥ 12 months	1015 (52.0)	33 (64.7)	806 (53.9)	137 (43.9)	39 (41.1)
Bottle use > 15 months	624 (32.0)	11 (21.6)	447 (29.9)	124 (39.7)	42 (44.2)
Daily cow’s milk intake > 2 cups	512 (26.2)	11 (21.6)	376 (25.2)	98 (31.4)	27 (28.4)
Meat consumption in last 3 days	1883 (96.4)	49 (96.1)	1439 (96.3)	302 (96.8)	93 (97.9)
**Laboratory characteristics**					
Serum ferritin (µg/L)	27.4 (18.7)	29.5 (19.7)	28.1 (19.6)	24.8 (13.8)	22.8 (16.0)
Hemoglobin (g/L) ^2^	118.1 (8.7)	113.5 (7.7)	118.0 (8.8)	119.4 (8.4)	118.9 (7.4)
Iron deficiency	269 (13.8)	8 (15.7)	197 (13.2)	44 (14.1)	20 (21.1)
Iron-deficiency anemia ^2^	92 (5.4)	2 (4.7)	70 (5.4)	15 (5.4)	5 (6.1)
CRP (mg/L)	0.6 (1.0)	0.6 (0.9)	0.6 (1.0)	0.6 (0.8)	0.6 (0.9)
≤1.0	1648 (84.4)	45 (88.2)	1257 (84.1)	265 (84.9)	81 (85.3)
>1.0 to ≤5.0	305 (15.6)	6 (11.8)	238 (15.9)	47 (15.1)	14 (14.7)

Data regarding baseline characteristics are presented as mean (SD) or N (%). zBMI indicates body mass index z score; CRP indicates C-reactive protein. ^1^ Based on the final analytic sample: data are for one imputed data set. ^2^ Based on the 1711 children with complete data for IDA (weight category: underweight (zBMI < −2), 43; normal weight (−2 ≤ zBMI ≤ 1), 1309; at-risk-of-overweight (1 < zBMI ≤ 2), 277; overweight/obese (zBMI > 2), 82). ^3^ Maternal ethnicity: European includes Western European, Eastern European, and Australian or New Zealander; Non-European includes East Asian, Southeast Asian, South Asian, West Asian, African, Caribbean, Latin American, North American Indigenous, and Mixed (2 or more ethnic groups).

**Table 4 nutrients-15-03635-t004:** Multivariable linear regression models for the association of zBMI and weight category with median serum ferritin (µg/L) in children aged 12–29 months (*n* = 1953) ^1^.

Variable				Change in Median Serum Ferritin		
	β (log)	95% CI	Change in Serum Ferritin, %	µg/L	95% CI	*p*-Value
**BMI as a continuous variable**						
zBMI	−0.05	−0.08, −0.03	−5.25	−1.31	−1.93, −0.68	<0.001 †
**BMI categorized into weight categories**						
Weight category						
Underweight (zBMI < −2)	0.08	−0.10, 0.25	7.82	1.95	−2.28, 6.97	0.39
Normal weight (−2 ≤ zBMI ≤ 1)	–	–	–	–	–	–
At-risk-of-overweight (1 < zBMI ≤ 2)	−0.06	−0.13, 0.02	−5.64	−1.41	−3.14, 0.46	0.14
Overweight/obese (zBMI > 2)	−0.21	−0.34, −0.08	−19.26	−4.81	−7.27, −2.02	0.001 †

CI indicates confidence interval; zBMI indicates body mass index z score. Negative values for change in median serum ferritin indicate a decrease in median serum ferritin. ^1^ All models adjusted for prespecified covariates including age, sex, birthweight, family income, infant feeding in the 1st year of life, breastfeeding duration, prolonged bottle use, daily cow’s milk intake, C-reactive protein (CRP) level. Child age was transformed using restricted cubic spline transformation with 3 knots to correspond to age at scheduled visits (15, 18, and 24 months). † Statistically significant findings as *p* < 0.05.

**Table 5 nutrients-15-03635-t005:** Multivariable logistic regression models for the association of zBMI and weight category with iron deficiency (serum ferritin < 12 µg/L) in children aged 12–29 months (*n* = 1953) ^1^.

Variable	Iron Deficiency		
	β (95% CI)	OR (95% CI)	*p*-Value
**BMI as a continuous variable**			
zBMI	0.13 (0.004, 0.26)	1.14 (1.00, 1.30)	0.04 †
**BMI categorized into weight categories**			
Weight category			
Underweight (zBMI < −2)	−0.01 (−0.83, 0.80)	0.98 (0.44, 2.23)	0.97
Normal weight (−2 ≤ zBMI ≤ 1)	–	1.00	–
At-risk-of-overweight (1 < zBMI ≤ 2)	0.15 (−0.23, 0.53)	1.16 (0.80, 1.70)	0.44
Overweight/obese (zBMI > 2)	0.76 (0.20, 1.33)	2.15 (1.22, 3.78)	0.008 †

OR indicates odds ratio; CI indicates confidence interval; zBMI indicates body mass index z score. ^1^ All models adjusted for prespecified covariates including age, sex, birthweight, family income, infant feeding in the 1st year of life, breastfeeding duration, prolonged bottle use, daily cow’s milk intake, C-reactive protein (CRP) level. Child age was transformed using restricted cubic spline transformation with 3 knots to correspond to age at scheduled visits (15, 18, and 24 months). † Statistically significant findings as *p* < 0.05.

**Table 6 nutrients-15-03635-t006:** Multivariable linear regression model for the association between weight category and serum ferritin (*n* = 1953) ^1^.

Variable				Change in Median Serum Ferritin		
	β (log)	95% CI	Change in Serum Ferritin, %	µg/L	95% CI	*p*-Value
Weight category						
Underweight (zBMI < −2)	0.08	−0.10, 0.25	7.82	1.95	−2.28, 6.97	0.39
Normal weight (−2 ≤ zBMI ≤ 1)	–	–	–	–	–	–
At-risk-of-overweight (1 < zBMI ≤ 2)	−0.06	−0.13, 0.02	−5.64	−1.41	−3.14, 0.46	0.14
Overweight/obese (zBMI > 2)	−0.21	−0.34, −0.08	−19.26	−4.81	−7.27, −2.02	0.001 †
Child sex, female	0.09	0.04, 0.15	9.85	2.46	1.01, 4.00	<0.001 †
Child age, months ^2^						
12–15 months	−0.06	−0.10, −0.03	−6.23	−1.56	−2.30, −0.79	<0.001 †
15–18 months	0.001	−0.03, 0.04	0.14	0.04	−0.82, 0.92	0.94
18–24 months	−0.0005	−0.01, 0.01	−0.04	−0.01	−0.35, 0.33	0.95
24–29 months	0.04	−0.002, 0.07	3.61	0.90	−0.04, 1.88	0.06
Birthweight, kg	0.08	0.03, 0.13	8.36	2.09	0.82, 3.42	0.001 †
CRP level	0.07	0.05, 0.10	7.55	1.89	1.15, 2.64	<0.001 †
Family income (CAD)						
Less than 40,000 CAD	−0.16	−0.27, −0.05	−14.64	−3.66	−5.91, −1.15	0.005 †
40,000–79,999 CAD	0.08	−0.005, 0.17	8.83	2.21	−0.11, 4.74	0.06
80,000–149,999 CAD	−0.01	−0.08, 0.06	−1.39	−0.35	−1.99, 1.42	0.69
150,000+ CAD	–	–	–	–	–	–
Infant feeding in first year of life						
Mostly breastfed	–	–	–	–	–	–
Breast milk and formula equally	0.20	0.12, 0.29	22.67	5.67	3.15, 8.41	<0.001 †
Mostly formula-fed	0.29	0.20, 0.39	34.31	8.58	5.52, 11.95	<0.001 †
Breastfeeding duration ≥ 12 months	−0.17	−0.25, −0.09	−15.96	−3.99	−5.59, −2.26	<0.001 †
Bottle use > 15 months	−0.07	−0.15, 0.01	−6.49	−1.62	−3.38, 0.27	0.09
Cow’s milk intake > 2 cups/d	−0.17	−0.24, −0.10	−15.53	−3.88	−5.36, −2.29	<0.001 †

CI indicates confidence interval; zBMI indicates body mass index z score; CRP indicates C-reactive protein. Negative values for change in median serum ferritin indicate a decrease in median serum ferritin. ^1^ All models adjusted for prespecified covariates including age, sex, birthweight, family income, infant feeding in the 1st year of life, breastfeeding duration, prolonged bottle use, daily cow’s milk intake, CRP level. ^2^ Child age was transformed using restricted cubic spline transformation with 3 knots to correspond to the age at scheduled visits (15, 18, and 24 months). † Statistically significant findings as *p* < 0.05.

**Table 7 nutrients-15-03635-t007:** Multivariable logistic regression model for the association between weight category and iron deficiency (serum ferritin < 12 µg/L) (*n* = 1953) ^1^.

Variable	Iron Deficiency		
	β (95% CI)	OR (95% CI)	*p*-Value
Weight category			
Underweight (zBMI < −2)	−0.01 (−0.83, 0.80)	0.98 (0.44, 2.23)	0.97
Normal weight (−2 ≤ zBMI ≤ 1)	–	1.00	–
At-risk-of-overweight (1 < zBMI ≤ 2)	0.15 (−0.23, 0.53)	1.16 (0.80, 1.70)	0.44
Overweight/obese (zBMI > 2)	0.76 (0.20, 1.33)	2.15 (1.22, 3.78)	0.008 †
Child sex, female	−0.15 (−0.42, 0.12)	0.86 (0.65, 1.13)	0.28
Child age, months ^2^			
12–15	0.18 (0.01, 0.35)	1.20 (1.01, 1.41)	0.03†
15–18	−0.02 (−0.19, 0.14)	0.98 (0.83, 1.15)	0.78
18–24	−0.02 (−0.09, 0.04)	0.98 (0.92, 1.05)	0.54
24–29	−0.14 (−0.35, 0.08)	0.87 (0.70, 1.08)	0.21
Birthweight, kg	−0.47 (−0.71, −0.23)	0.63 (0.49, 0.80)	<0.001 †
CRP level	−0.14 (−0.31, 0.03)	0.87 (0.73, 1.03)	0.10
Family income (CAD)			
Less than 40,000 CAD	1.05 (0.59, 1.51)	2.85 (1.80, 4.52)	<0.001 †
40,000–79,999 CAD	−0.45 (−0.98, 0.08)	0.64 (0.38, 1.09)	0.10
80,000–149,999 CAD	0.20 (−0.16, 0.56)	1.22 (0.86, 1.75)	0.27
150,000+ CAD	–	1.00	–
Infant feeding first year of life			
Mostly breastfed	–	1.00	–
Breast milk and formula equally	−0.81 (−1.28, −0.34)	0.44 (0.28, 0.71)	<0.001 †
Mostly formula-fed	−0.97 (−1.53, −0.40)	0.38 (0.22, 0.67)	<0.001 †
Breastfeeding duration ≥ 12 months	0.63 (0.22, 1.03)	1.87 (1.25, 2.81)	0.003 †
Bottle use > 15 months	0.16 (−0.22, 0.53)	1.17 (0.81, 1.70)	0.41
Cow’s milk intake > 2 cups/d	0.61 (0.28, 0.93)	1.84 (1.33, 2.54)	<0.001 †

OR indicates odds ratio; CI indicates confidence interval; zBMI indicates body mass index z score; CRP indicates C-reactive protein. ^1^ All models adjusted for prespecified covariates including age, sex, birthweight, family income, infant feeding in the 1st year of life, breastfeeding duration, prolonged bottle use, daily cow’s milk intake, CRP level. ^2^ Child age was transformed using restricted cubic spline transformation with 3 knots to correspond to the age at scheduled visits (15, 18, and 24 months). † Statistically significant findings as *p* < 0.05.

## Data Availability

The data presented in this study are available on request from the corresponding author. The data are not publicly available due to privacy.

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
