# Peer review of "Examining the Double Burden of Underweight, Overweight/Obesity and Iron Deficiency among Young Children in a Canadian Primary Care Setting"

_nutrients, 2023, doi:10.3390/nu15163635_

Round 1

Reviewer 1 Report

The study of Canadian authors, which showed an association between higher BMI and iron deficiency in young children, was designed extremely carefully. The manuscript itself is also well prepared, all elements are clearly defined. The aim of the work is clear, and the methodology and results allowed for its implementation and drawing legitimate conclusions from the research.

However, it would be interesting to discuss whether other markers of iron metabolism might be useful in evaluating the relationship between body weight and iron deficiency in children. I suggest these two publications: Cancers (Basel). 2023;15(4):1041; Children (Basel). 2023;10(5):870.

Reviewer 2 Report

The manuscript assessed the double burden of malnutrition in a developed country context and found that overweight/obesity was associated with iron deficiency. Overall, the manuscript reads well. I have a few minor comments/suggestions.

As being underweight was not associated with iron deficiency relative to normal weights, it seems likely that the relationship between continuous BMI and serum ferritin was not linear. The authors may want to address this.

About 8% (n=208) children had missing data on BMI and were excluded from the analysis. It would strengthen the manuscript if the characteristics of children with and without missing data on BMI are comparable.

May consider move Tables 5 and 6 to appendix if assessing risk factors for iron deficiency was not the key aim of the study because the information on weight categories was repetitive.
